# Intelligent auxiliary system for music performance under edge computing and long short-term recurrent neural networks

Yi Wang *

KU School of Music, Lawrence, Kansas, United States of America

* YvonneWang@ku.edu

## Abstract

Music performance action generation can be applied in multiple real-world scenarios as a research hotspot in computer vision and cross-sequence analysis. However, the current generation methods of music performance actions have consistently ignored the connection between music and performance actions, resulting in a strong sense of separation between visual and auditory content. This paper first analyzes the attention mechanism, Recurrent Neural Network (RNN), and long and short-term RNN. The long and short-term RNN is suitable for sequence data with a strong temporal correlation. Based on this, the current learning method is improved. A new model that combines attention mechanisms and long and short-term RNN is proposed, which can generate performance actions based on music beat sequences. In addition, image description generative models with attention mechanisms are adopted technically. Combined with the RNN abstract structure that does not consider recursion, the abstract network structure of RNN-Long Short-Term Memory (LSTM) is optimized. Through music beat recognition and dance movement extraction technology, data resources are allocated and adjusted in the edge server architecture. The metric for experimental results and evaluation is the model loss function value. The superiority of the proposed model is mainly reflected in the high accuracy and low consumption rate of dance movement recognition. The experimental results show that the result of the loss function of the model is at least 0.00026, and the video effect is the best when the number of layers of the LSTM module in the model is 3, the node value is 256, and the Lookback value is 15. The new model can generate harmonious and prosperous performance action sequences based on ensuring the stability of performance action generation compared with the other three models of cross-domain sequence analysis. The new model has an excellent performance in combining music and performance actions. This paper has practical reference value for promoting the application of edge computing technology in intelligent auxiliary systems for music performance.

**Data Availability Statement:** All relevant data are within the manuscript and its Supporting information files.

**Funding:** The author received no specific funding for this work.

**Competing interests:** The authors have declared that no competing interests exist.

## 1. Introduction

Music and dance have always been inseparable arts because dance accompanied by music can satisfy the double enjoyment of the audience's vision and hearing. With the rapid development of Deep Learning (DL) technology and artificial intelligence, the realization of computer-generated dance performance actions through music is a hot research topic. Although there is no one-to-one correspondence between performance actions and musical rhythms, there is a robust correlation between the rhythms of the two. This strong correlation provides critical clues for cross-domain sequence analysis of music and performance actions [1, 2]. Although the current research on music performance action generation has achieved specific results relying on DL, the algorithm's shortcomings limit the practical application of music performance action generation. Apart from algorithmic issues, there is currently a lack of comprehensive analysis on the extraction of musical features. There is no suitable dance synthesis method. Therefore, a significant and challenging point of current related research is to reasonably analyze the characteristics of music and dance and generate a corresponding mapping relationship between them. At present, the corresponding research results have been abundant. Alemi (2017) showed preliminary results on GrooveNet. GrooveNet was a generative system that learned to synthesize dance moves for a given soundtrack in real-time. The application designed by GrooveNet was a public interactive installation. Viewers could provide their music to interact with the avatar. Artificially trained neural networks were mainly studied, especially factor-conditioned restricted Boltzmann machines and Recurrent Neural Networks (RNNs). Recordings of dance moves for this project were captured on four small datasets of synchronized music and actions. Preliminary results indicated that it was possible to train on this small dataset to generate dance moves [3]. Zhuang (2022) especially proposed a novel autoregressive generative model DanceNet to make the synthesized actions consistent with the music's style, rhythm, and melody. The music's style, rhythm, and melody are used as control signals to generate 3D dance moves with high realism and variety. Dilated convolutions were proposed to improve the effect due to the spatiotemporal complexity of dance. Gated activation units and separable convolutions were used to enhance the fusion of motion features and control signals. Besides, a high-quality dataset was constructed by synchronizing music dances by several professional dancers to improve the performance of the model [4]. Srinivasu et al. (2021) researched the relationship between DL neural networks and skin disease classification diagnosis through the development and use of Convolutional Neural Networks (CNNs) and visual geometry groups. The results showed that the proposed system could help general practitioners effectively diagnose skin conditions, reducing further complications and morbidity [5]. Chandra et al. (2022) studied the application of DL technology based on Long Short-Term Memory (LSTM) model in the prediction of the COVID-19. They re-examined the epidemic changes through reliable data sources and innovative prediction models. The results indicated that the accuracy of model prediction could be greatly improved by DL through LSTM model [6]. In addition, the problems existing in the current recognition of music and dance movements are analyzed. It can be found that there is a certain gap between the current technology and the expected recognition accuracy of the model. The current model's music intelligent assistance level is low, which cannot achieve the effective combination of performance action and music beat.

The main contribution of this paper is to analyze and optimize the relationship between music and dance performance movements using the attention mechanism model. The main motivation of this paper is to design and apply intelligent auxiliary systems for music performance through edge computing and long short-term RNNs to improve the performance level of music and dance movements. The overall structure is shown as follows. Section 1 is the

background of the discussion on music performance and dance movement recognition. Section 2 introduces the attention mechanism and neural network models. Section 3 analyzes the training and test results of the model through the establishment and experiment of network model. Section 4 draws research conclusions through systematic induction and summary. The research has practical reference value for promoting the intelligent level of music performance.

## 2. Materials and methods

### 2.1 The principle of attention mechanism

The attention mechanism originates from the study of human vision. It is first discovered in cognitive science. Humans often only selectively focus on the essential local information of all information to overcome the bottleneck of information processing while ignoring other visible details. The attention mechanism decides which part of the input needs to focus on and allocates limited information processing resources to the critical components. The attention mechanism is extended to the learning of the NN model. Its principle is to divide the image to be processed into different regions and assign weights to the other areas. According to the size of the assigned weights, much attention can be devoted to the critical areas of the image, and the attention to useless information in other areas can be reduced. On the one hand, it can obtain detailed information on critical areas. On the other hand, it also reduces the complexity of processing, making the learning and training of NN models flexible. The image description generation model that introduces the attention mechanism is shown in Fig 1.

From Fig 1, the encoder of this model uses a CNN to extract the features of the image. The decoder uses an RNN. It converts the previously extracted image features into textual information matching the image content. In this process, the focus area of the image is determined according to the weight of the attention. The attention mechanism is adopted here for feature extraction. The resulting features are typical of sequence data, so what follows is a NN suitable for processing sequence data.

### 2.2 RNN

The earliest research on RNN started in the 1980s and 1990s. RNN has become one of the most common algorithms in DL. RNN takes sequence data as input. The input is recursively carried out in the evolution direction of the sequence, and all the recurrent units are connected in a chain. The primary function is to perform repetitive operations on a set of sequential data inputs [7]. There is no absolute correlation between the previous input and the following input of the CNN. While it is possible to make all inputs relatively independent, the outputs can also be completely uncorrelated. Therefore, the connection before and after cannot be considered in processing time-series data. Each layer of RNN can memorize all previous unit information, indicating that it can review infinitely. In practice, RNN can only remember the information of a few steps back, so most scholars improve the memory ability of previous information by improving the structure of RNN [8]. Fig 2 displays the simple structure of RNN.

In Fig 2, the vectors X, S, and O represent the values of the input, hidden, and output layers. The matrix U and matrix V represent the calculation weight of the data from the input layer to the hidden layer and the weight of the hidden layer to the output layer. If the cycle layer is not considered, the specific structure analysis is shown in Fig 3.

From Fig 3, this is a simple, fully connected NN. X as the input layer is a three-dimensional vector, and U is the parameter matrix from the input layer to the hidden layer. The dimension in the figure is 3×4, and S is the vector of the hidden layer. In the figure above, the dimension is 4×2, and V is the parameter matrix from the hidden layer to the output layer. O is the vector

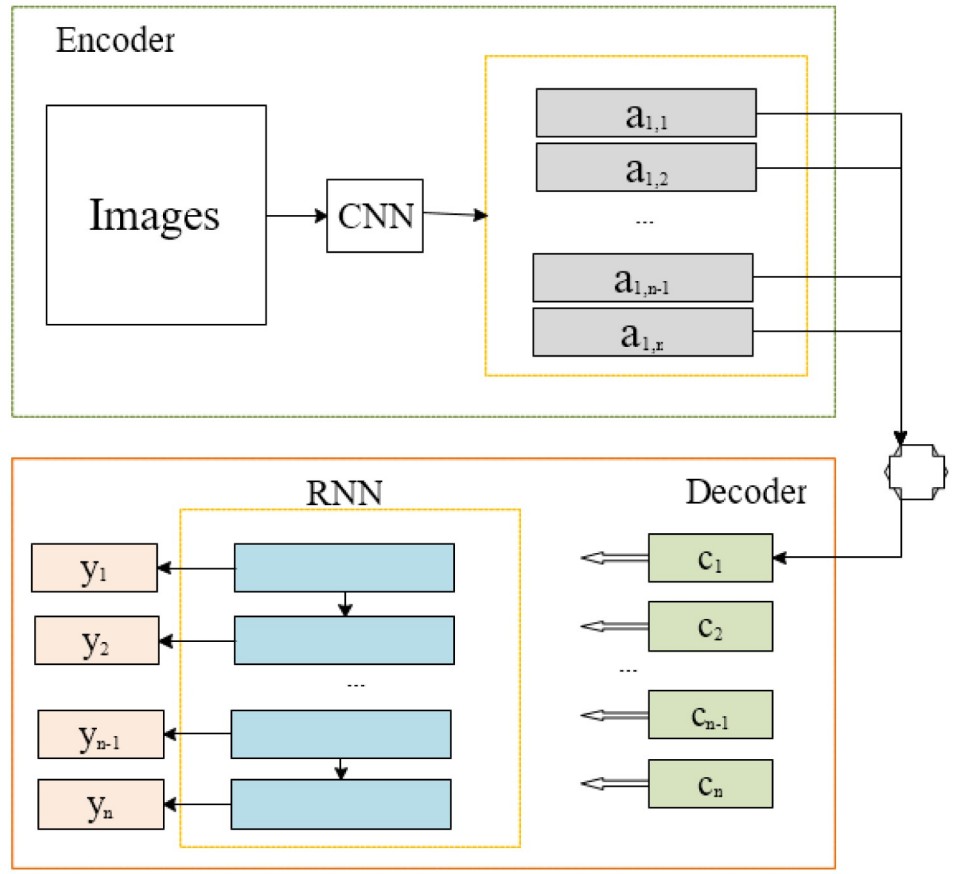

**Fig 1. Image description generation model with the attention mechanism.**

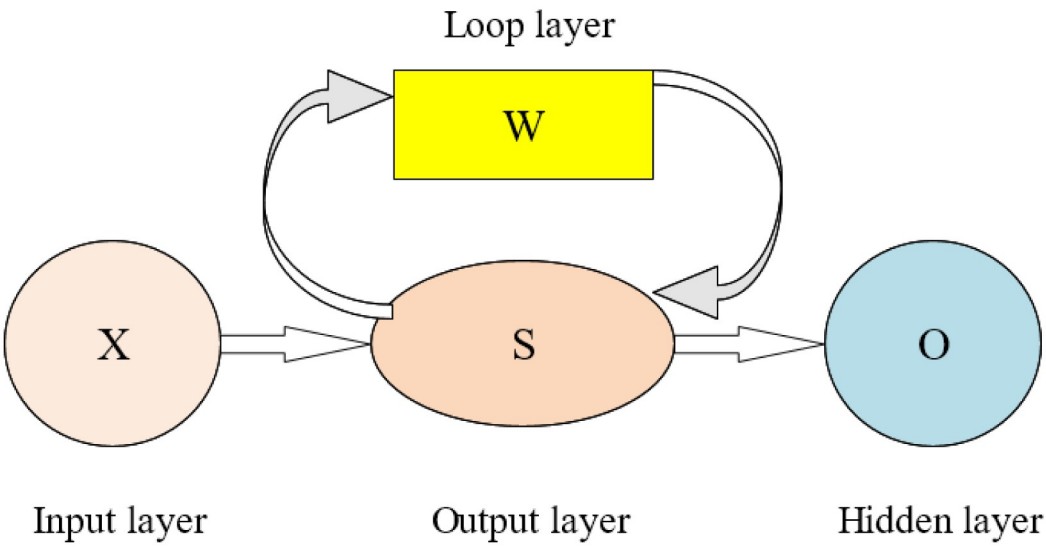

**Fig 2. RNN abstract structure diagram.**

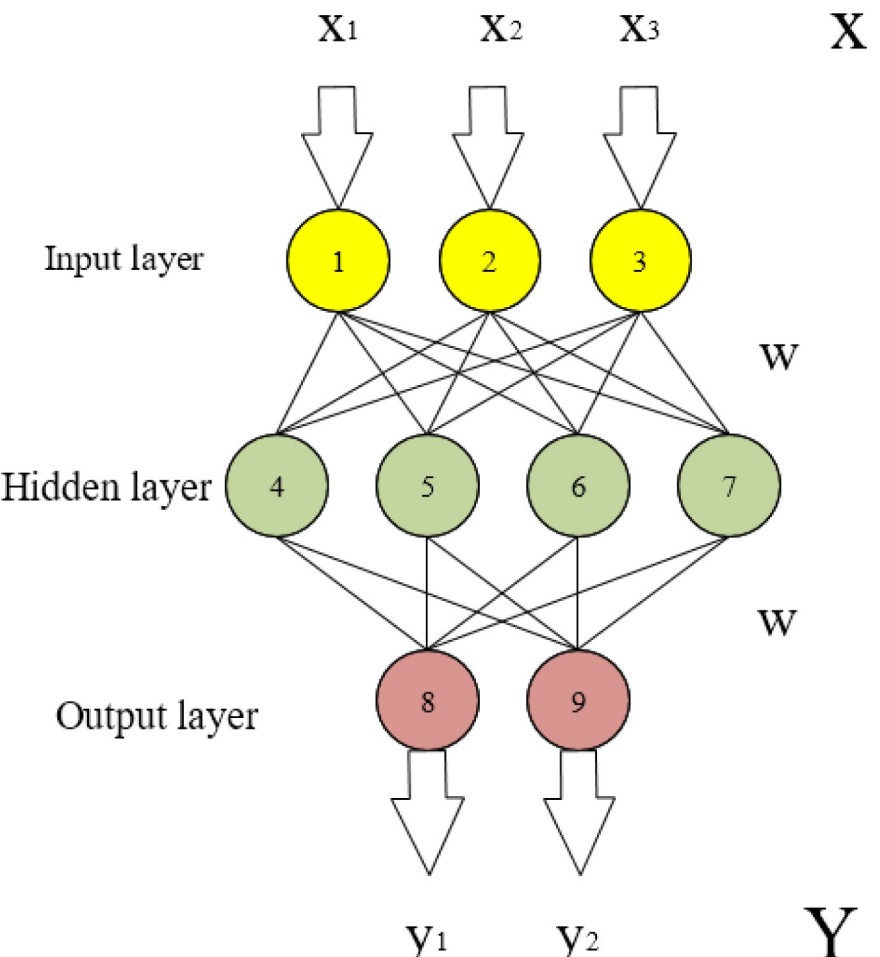

**Fig 3. The specific structure of the RNN network without considering the recurrent layer.**

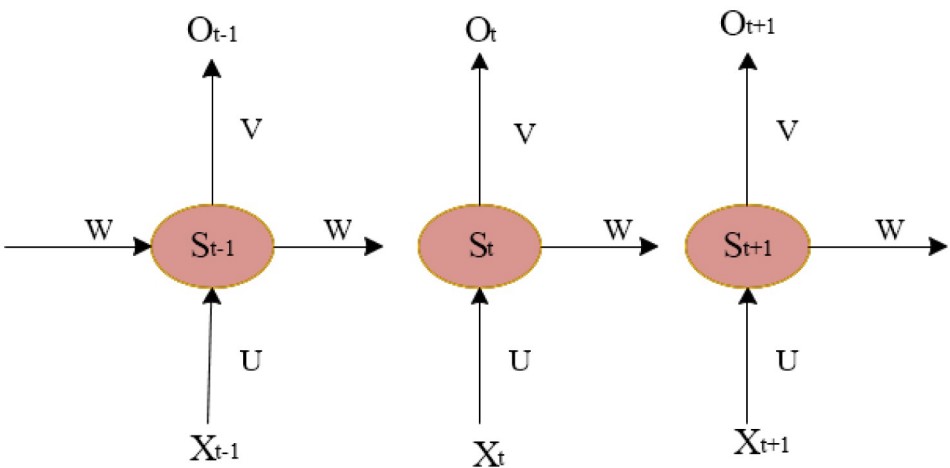

**Fig 4. Expanded RNN abstract structure diagram according to the timeline.**

of the output layer. In the above figure, the dimension is 2 [9, 10]. Fig 2 is expanded according to the timeline. Then, Fig 4 shows its structure.

From Fig 4, when the input value $X_t$ is received at $t$, the value of the hidden layer is $S_t$, and the output value is $O_t$. Therefore, RNN can solve the principle of sequence problems. It can remember every moment of information. The hidden layer $S_t$ at each moment is determined by the input layer $X_t$ at that moment and the hidden layer $S_{t-1}$ at the previous moment [11]. The way each layer of the RNN is calculated is as follows.

$$O_t = g\left(V_{s_t}\right) \tag{1}$$

$$S_t = f(UX_t + WS_{t-1}) \tag{2}$$

The RNN output layer can be calculated according to Eq (1). Ot represents the output at time t. The output layer is fully connected. The hidden layer can be calculated according to Eq (2). St represents the value of the hidden layer at time t, while V and W represent the corresponding weight matrix [12]. Eq (2) is substituted into Eq (1) repeatedly, and there are:

$$O_t = Vf(UX_t + WS_{t-1}) \tag{3}$$

$$O_t = Vf(UX_t + Wf(UX_{t-1} + WS_{t-2})) \tag{4}$$

$$O_t = Vf(UX_t + Wf(UX_{t-1} + Wf(UX_{t-2} + WS_{t-3}))) \tag{5}$$

$$O_t = Vf(UX_t + Wf(UX_{t-1} + Wf(UX_{t-2} + Wf(UX_{t-3}) + \cdots)))) \tag{6}$$

In Eqs (3)–(6), finding the value of the output layer 0$t$ of RNN needs to consider the influence of all the previous input layers $X$ $Xt$, $Xt-1$, $Xt-2$, $Xt-3$,.. … .. This is exactly why RNNs are suitable for time series data. However, the scale of model parameters becomes extremely large as RNN increases with the number of network layers. It leads to the problem of gradient explosion and disappearance during training. As a result, the memory of the current layer to the distant previous layer decays rapidly [13]. The problem of gradient disappearance can be solved by adjusting the initialized weight, changing the activation function, or using a variant of RNN. The most suitable one is the long and short-term RNN.

## 2.3 Long and short-term RNN

The long and short-term RNN can alleviate the defects of CNN to a certain extent compared with the traditional RNN explained before. Fig 5 demonstrates the simplified schematic diagram of its principle.

From Fig 5, a new unit state c is added when the RNN has only one hidden state, h. h is sensitive to short-term input, and c can save long-term state. Fig 5 is expanded in the time dimension, and its structure is shown in Fig 6.

In Fig 6, xt represents the input parameters at time t. ht-1 represents the input parameters of the previous time t-1. ct-1 represents the previous cell state. ht-1 represents the output value at the previous moment. The core of the long and short-term RNN controls the newly added unit state c [14]. The idea is the gating algorithm. The RNN is given the ability to control the accumulation of its internal information through gating units. In learning, it can not only master long-distance dependencies but also selectively forget information to prevent overload through three control gates [15]. The framework structure of the proposed model is shown in Fig 7.

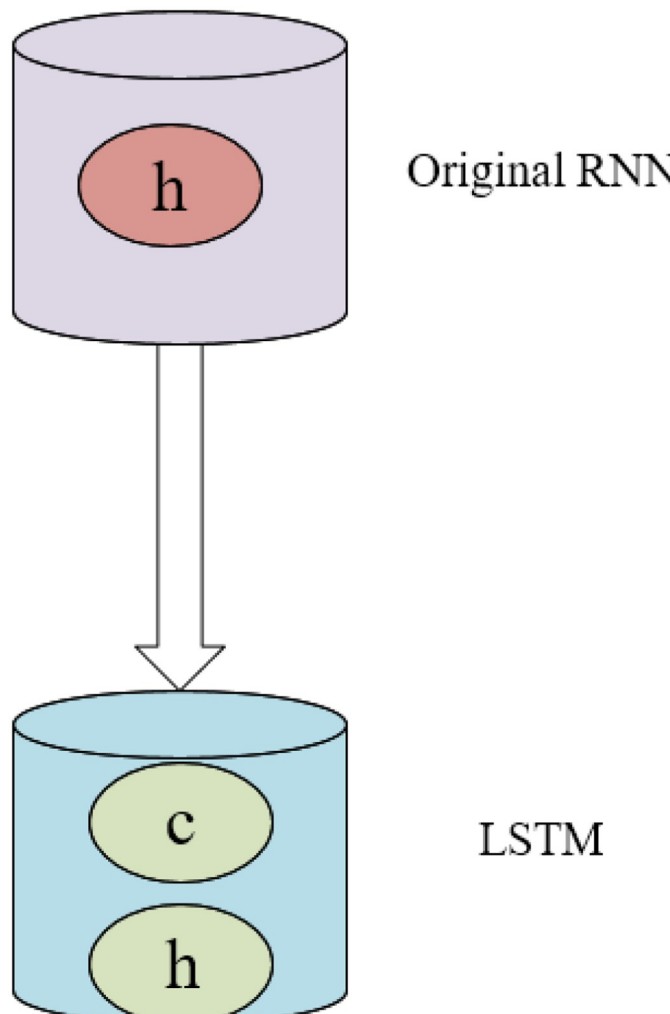

**Fig 5. RNN-LSTM abstract schematic.**

In LSTM structure network, the functions of the three control gates are represented by switches. The three control gates are: forget gate, input gate, and output gate. The forget gate determines the information retained by the cell state ct-1 at the previous time t-1 to the current time ct. The input gate determines the amount of information from the input xt at present t to the cell state ct. The output gate determines the amount of information from the long-term cell state ct to the output value ht of the model at this moment. The function of gating is to allow information to pass selectively. The output range of the activation function Sigmoid is from zero to one to define the degree of passing through the gate. Zero represents that all information is not accessible, while one means that all information is fully accessible [16]. The specific calculation process is as follows.

$$f_t = \sigma\left(w_{f_t}x_t + w_{f_h}h_{t-1} + b_f\right) \tag{7}$$

$$i_t = \sigma\left(w_{i_x}x_t + w_{ih}h_{t-1} + b_i\right) \tag{8}$$

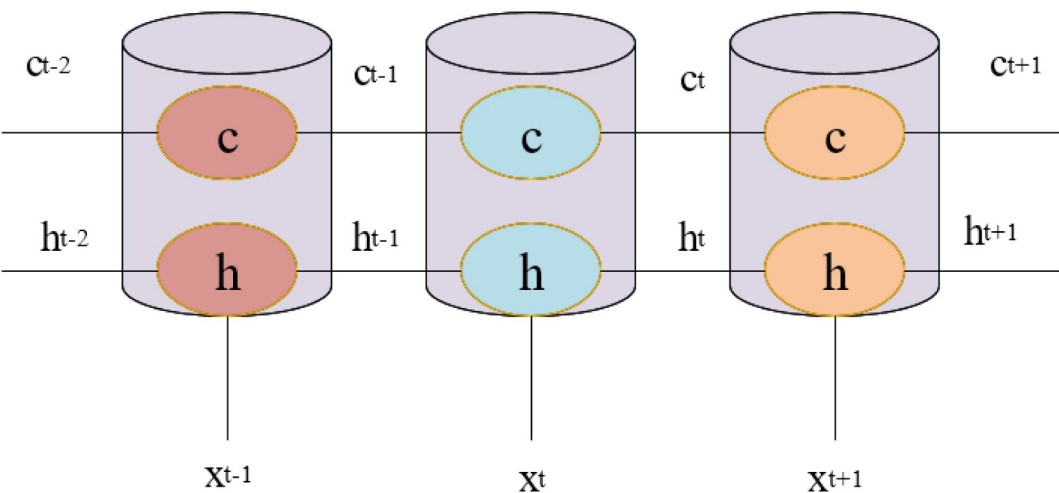

**Fig 6. Expanded view of LSTM structure.**

$$\tilde{c}_t = tanh(w_{cx}x_t + w_{ch}h_{t-1} + b_c) \tag{9}$$

The forget and input gates can be calculated according to Eqs (7) and (8). Eq (9) represents the calculation when the state is updated. c represents the internal state. h represents the system state. b represents the weight. f, i, and o each represent a forget gate, an input gate, and an output gate. σ represents the Sigmoid function, and Tanh represents the hyperbolic tangent

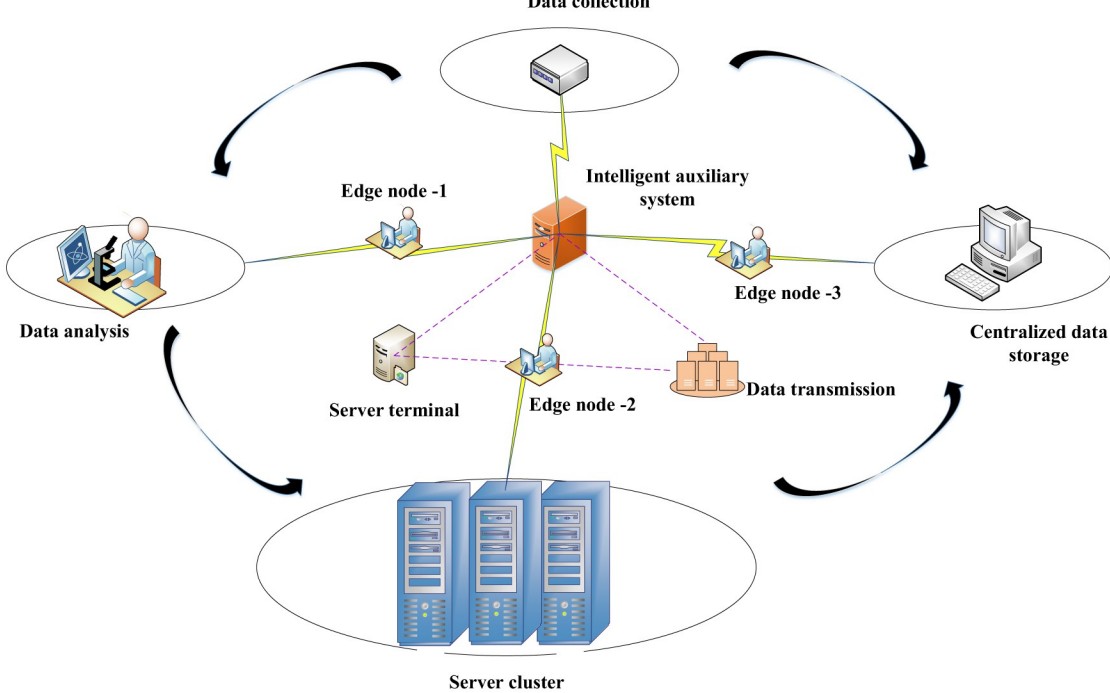

**Fig 7. The framework structure diagram of the proposed intelligent assistance system.**

function. Two influencing factors of the current input unit state $\tilde{c}_t$ are the output of the previous unit state and the input given by the existing state network [17]. Besides, the unit state $\tilde{c}_t$ is obtained according to Eq (10).

$$c_t = f_t \circ c_{t-1} + i_t \circ \tilde{c}_t) \tag{10}$$

In Eq (10), ct-1 represents the unit's state at the previous moment. *ft* represents the forget gate. *it* represents the input at the current moment. The symbol ∘ represents the multiplication of the corresponding elements of two vectors. The current state memory $\tilde{c}_t$ in the long and short-term RNN can be combined with the previously accumulated memory *ct*−1 for a long time through the above operations. The calculation of the output gate is shown below.

$$c_t = \sigma(w_{ox}x_t + w_{oh}h_{t-1} + b_o) \tag{11}$$

The final output of the long and short-term RNN is jointly determined by the output gate and the unit state.

$$h_t = o_t \circ \tanh(c_t) \tag{12}$$

From the above calculations, the long and short-term RNN introduces the forget gate, which can retain useful information and forget other useless information, so it can save space to memorize the front data in the time series. The input gate is mainly to continuously input new information to iterate the forward timing information for efficient processing. The output gate will analyze the value of the information stored in the previous period. The final output is determined jointly by the output gate and the unit state. Long and short-term RNNs can be crucial in developing RNNs to generate variants [18].

## 2.4 Music-dance feature extraction

There are many features in both music and dance action information. Representative and strongly correlated music and dance features are selected here. The typical musical feature of beat is chosen to achieve the correlation between the two. The combination rule between strong and weak beats in music is reflected in the accent position in the music information. The rhythm can clearly express the mood of the music. For example, 2/4 time in music refers to one strong and one weak. The quarter note is a beat, with two beats per measure. There can be two-quarter notes representing the march's rhythm and liveliness. 3/4 beats are strong, weak, and weak, meaning swaying rhythm and emotional serenity. 4/4 beats are strong, weak, second strong, and weak, representing solemnity and sadness. The dance performance is also a series of actions according to the rhythm of the music [19]. The following are the music and dance feature extraction methods used here.

For music feature extraction, its overall structure is shown in Fig 8.

In the data preprocessing stage, sampling is performed using the sampling rate fs = 44.1kHz configuration. The result is used as a later network input. The audio signal is averaged. The purpose is to convert the stereo signal to a mono signal to reduce the amount of post-processing. After the signal is divided into frames and windowed in turn, the Fourier transform is performed to divide it into sound waves of different frequencies so that the neural network can learn later. In addition, the calculation of the power spectrum and the conversion of the Mel spectrum are performed. Mel spectral coefficients can describe the contour of the spectral envelope. The envelope peak position and height are essential features for beat discrimination. The final beat recognition result is displayed by the moment when the beat appears. The corresponding time indicates the time the beat point appears in a piece of music.

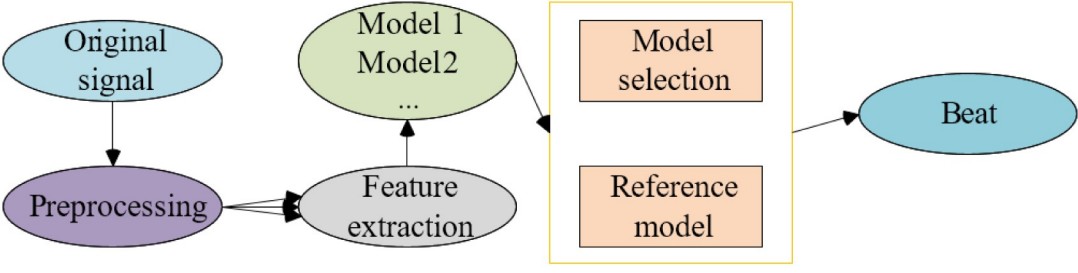

**Fig 8. Structure of music beat recognition.**

This paper regards the dancer's dance actions as a collection of different human body parts coordinate points to simplify the actions and obtain critical dance features. Therefore, the action information can be obtained by recognizing the human body posture of each frame image in the video so that the position information can be saved as the dance feature. Here, OpenPos, an open-source system for multi-person pose detection, is used for initial human pose estimation, which can detect the main parts of the human body, including limbs and faces. Feature extraction is revealed in Fig 9.

From Fig 9, extracting the Mel spectrogram requires adjusting the value of the number of frames. The goal is to have the same frame rate for both the music and dance features. After the OpenPose system processes the video, the 2D coordinate positions of 14 key points can be obtained, including the simulated robot dance actions in each frame, as shown in Fig 10.

Fig 10 shows the specific positions of the 14 limb critical points of the simulated robot when performing dance movements, and the information is normalized to limit it to the interval [-1,1]. On the one hand, subsequent data processing is convenient. On the other hand, the model runs with a fast convergence rate.

## 2.5 Edge server architecture and resource allocation strategy

Nodes need to be collected for data to solve the problems of long access time and limited network bandwidth when many users access. The system architecture of the edge server is built through the data source and the computing resource allocation strategy of the cloud computing center. The user's resource computing and storage tasks are assigned to different edge servers for processing through the task-processing environment built by the underlying operating system and edge service functions. The general structure of edge computing built by users in the network is analyzed. The framework comprises three parts: the cloud computing layer,

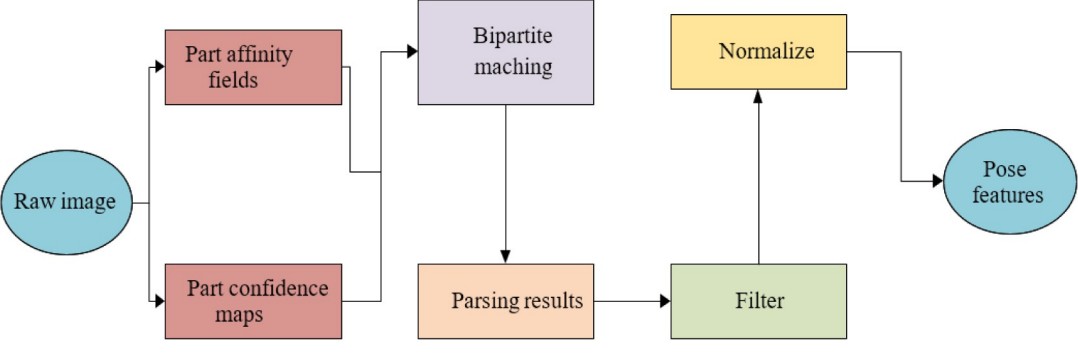

**Fig 9. Flow chart of dance action feature extraction.**

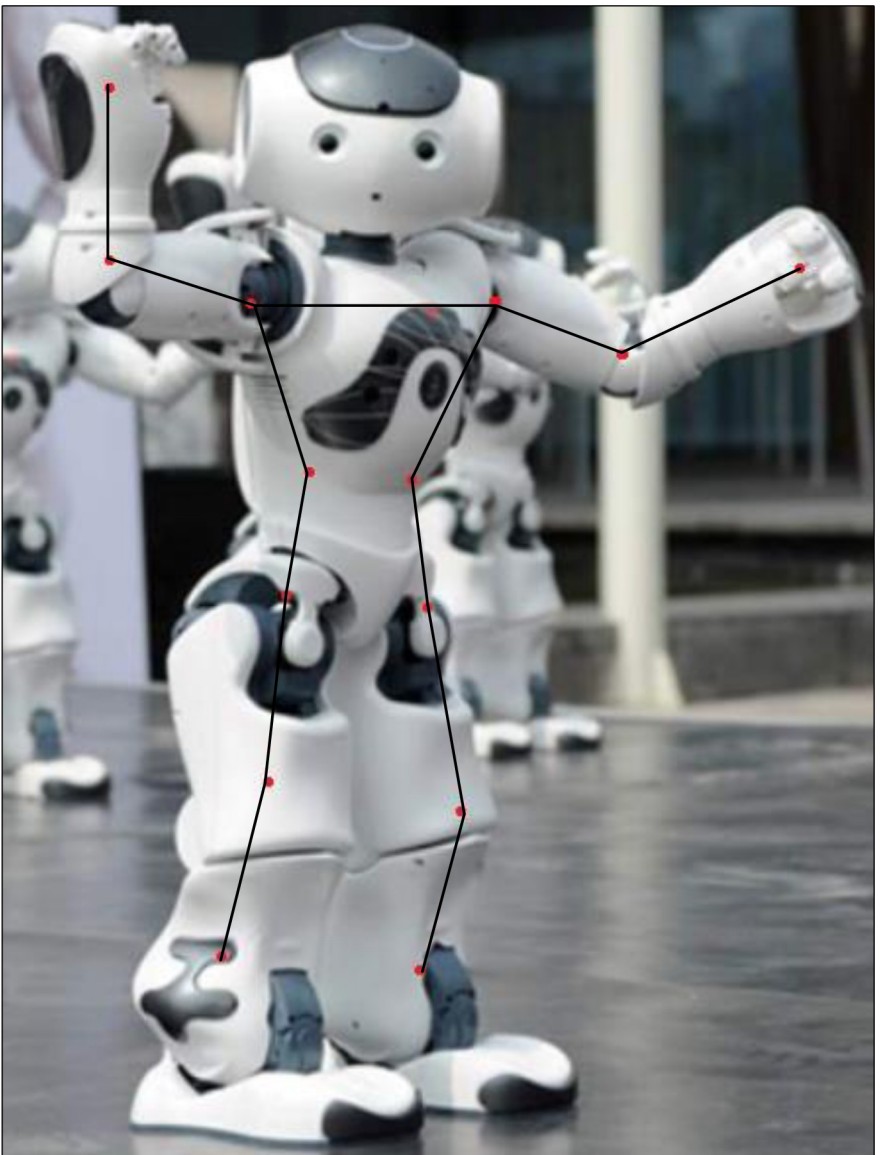

**Fig 10. 2D location map of 14 key points in robot dance action.**

edge layer, and field layer. On the monitoring platform of music and dance movements, musical features are jointly positioned through multiple microphones. Communication and data transmission between the individual system modules are connected using routers. The hardware structure of the established edge server system is shown in Fig 11.

## 2.6 The establishment of the network model

Music needs to be linked to dance actions, so the model must reflect a high degree of non-linearity between music and dance actions. The previous elaboration of the long and short-term RNN shows excellent and stable performance and training results for cross-domain sequence analysis. Therefore, this paper adopts an improved network model of long and short-term RNN fused with the attention mechanism, and its structure is shown in Fig 12.

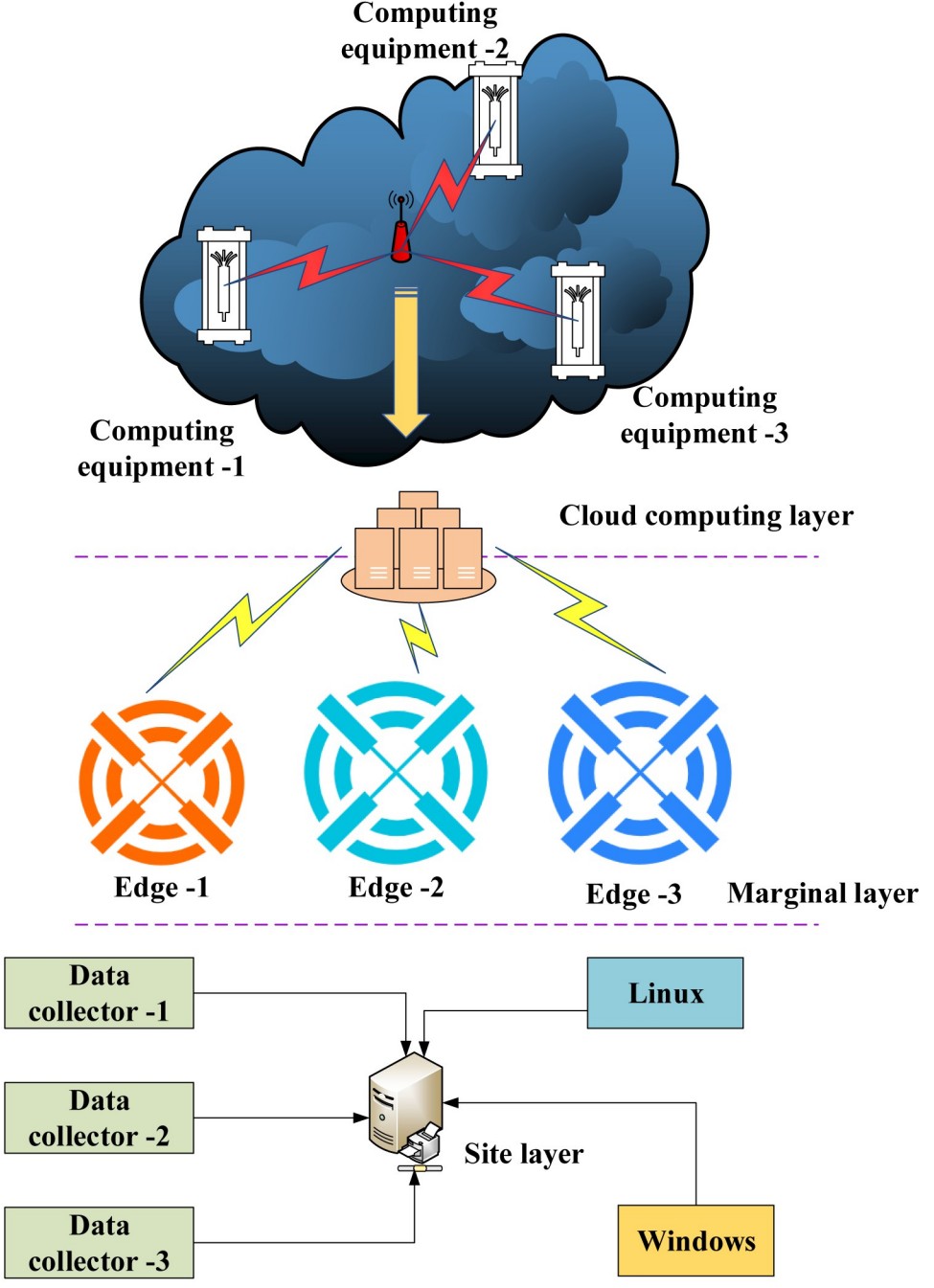

**Fig 11. Hardware structure diagram of the edge server system.**

From Fig 12, the NN structure is an improved model based on encoder-decoder long and short-term RNN. Besides, an attention mechanism is introduced to deal with the problem that the long and short-term RNN ignores the relationship between elements in the music sequence and the same semantic code of each frame of its output. On the one hand, all states in the encoder are preserved, and each element is assigned its weighted average. The semantic codes of each frame corresponding to the output are different. On the other hand, a new attention

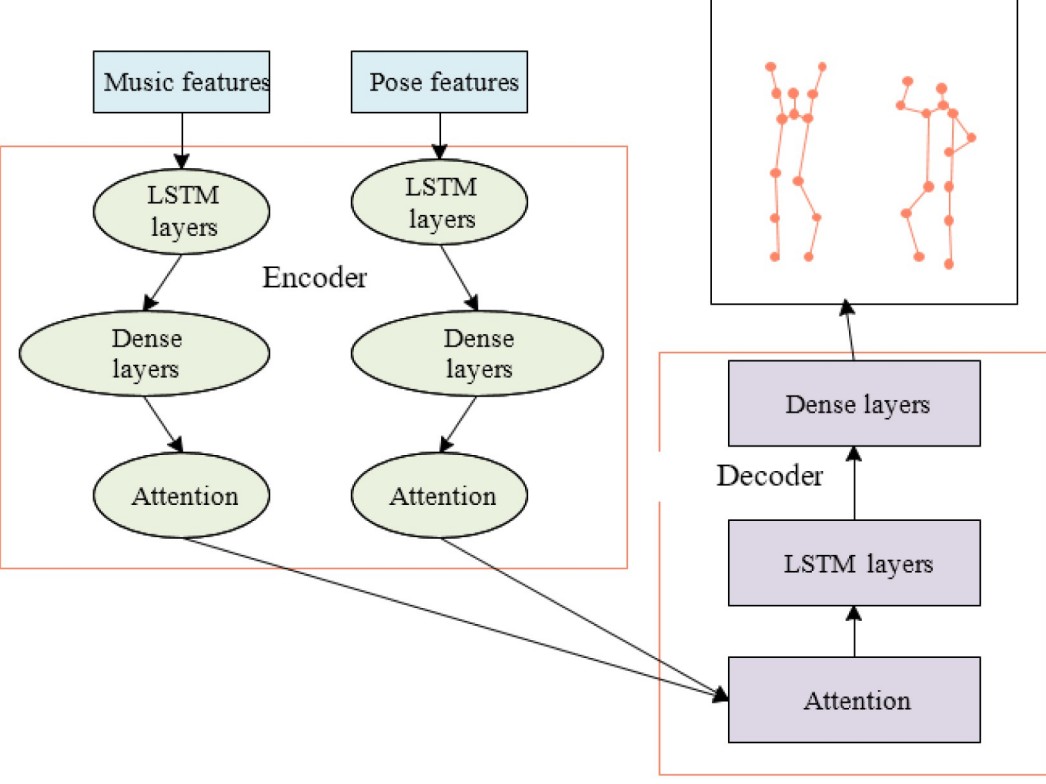

**Fig 12. Structure of the improved network model.**

mechanism can be added when the sequence is processed to obtain the relationship between the sequence elements. The new model is divided into three modules. The function of the LSTM module is to process the input information. The function of the Dense module is to output a sequence. The function of the Attention module is to change the decoding process of the decoder. The whole process is as follows. First, the beat and position features in OpenPose are extracted from the music and dance sequences. Then, the feature information of the two is input into the encoder network and goes through each layer in turn. The encoder result is fed into the decoder network and goes through each layer. When a new model is trained, music data is used as features and dance data as labels. The connection between music and dance actions is represented by adding the parameter Lookback. During training, the parameters of the NN are initialized first, and the forward propagation of the NN is performed. The network back-propagation calculation is implemented through the obtained loss function value, and the weight parameters of each layer are updated. When the number of iterations exceeds the maximum number of iterations, or the loss function is less than the error threshold, the training ends, and all training samples are saved for prediction.

## 3. Results and discussion

The data here are all from Music&Dance2019. This is a dataset of jazz dance genre music-dance pairs. A total of 60 different types of dance video data are selected therein. In addition, 48 of them are used to train the network model, and the other 12 videos are used to test the performance of the network model. The music-dance data to be trained and tested is preprocessed before being used as input data. In addition, in the process of establishing the network

model, the network structure characteristics of RNN and LSTM are analyzed. Input is 50, Tensor is 100, and Kernel size is 2*2. Stride 2 is also used. The type of activation function is Sigmoid. The proposed model executes 500 iterations with an initial learning rate of 0.005. The model saturates after 400 iterations. Combined with the relevant literature information [20], the specific settings of the network structure parameters are given in Table 1.

## 3.1 Experiments with different structural models

The experimental results of the loss function and visual effects are tested by changing the number of layers of the LSTM module, as demonstrated in Fig 13.

From Fig 13, when the number of layers increases from one to three, it is found that the loss function decreases significantly. When the number of layers is increased to five, the change in the loss value can be ignored. The LSTM proposed here finally chooses a three-layer network structure to reduce the over-fitting problem and training cost caused by too deep layers. Fig 13 reveals the experimental results of testing the loss function and visual effect by changing the number of nodes of the LSTM module. The loss function and visual effect change data of different modules of Loss are analyzed. Table 2 reveals the results.

From Fig 14, when a three-layer LSTM is used, the final results are different due to the different number of nodes in each layer. The loss value decreases significantly when the node value increases from 36 to 256. However, when the node value rises to 512, although the loss value decreases, the decrease is slight. The node value selected in the final model structure is 256 to reduce the volume of the model. The value of the Lookback will also affect the model's performance. The results are shown in Fig 15.

Fig 15 shows the trend of loss value under different Lookback values. It is found that when the lookback value is 1 or 70, the corresponding loss function curve changes very little, indicating that the number of music frames is too small or too long, which is not conducive to predicting dance actions. When the Lookback value is 15 and 30, the loss function value is the

**Table 1. Specific parameters setting of the network structure layer.**

| Parameter name | Parameter value |
| --- | --- |
| Number of layers | 5 |
| Units | 32 |
| Epoch | 4 |
| Batch size | 128 |
| Threshold value | 0.05 |
| Iterations | 500 |
| Input | 50 |
| Tensor | 100 |
| Kernel size | 2*2 |
| Stride | 2 |
| Activation function | Sigmoid |
| Learning rate | 0.005 |
| Training loss | 0.005 |
| Testing loss | 0.004 |
| Training accuracy | 0.74 |
| Testing accuracy | 0.76 |
| Training precision | 0.82 |
| Testing precision | 0.86 |

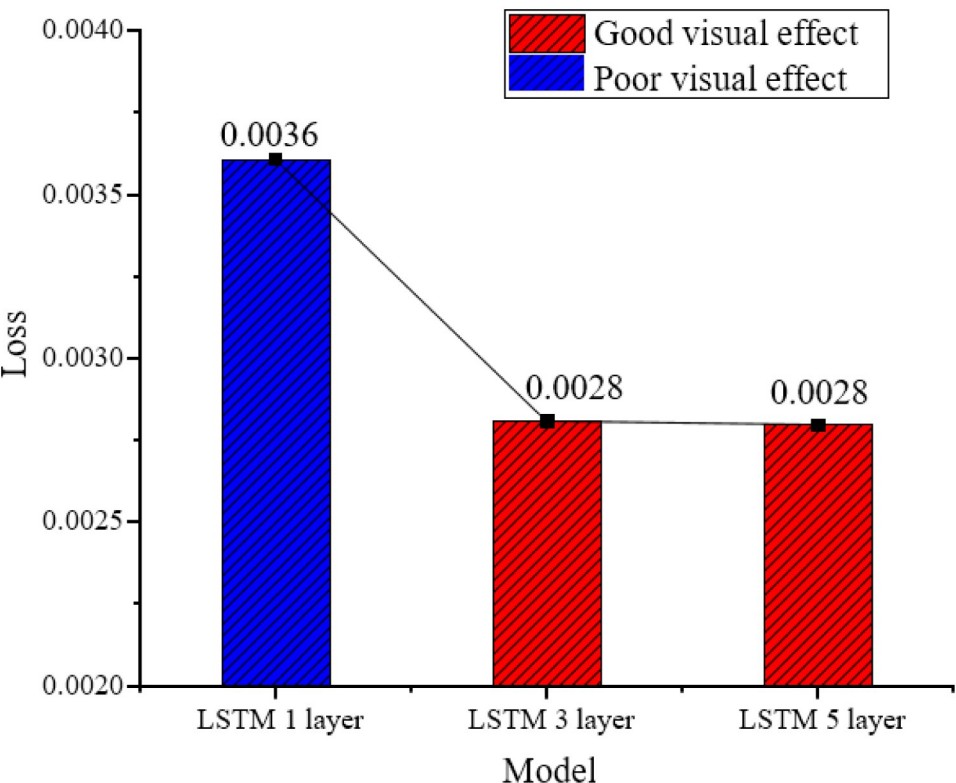

**Fig 13. Loss functions and visual effects with different layers of LSTM modules.**

smallest, and the change is noticeable. Finally, the Lookback value is set to 15 to strengthen the connection between music and dance actions.

## 3.2 Comparison between different cross-domain sequence models

The new network model proposed here is compared with other cross-domain sequence analysis models. The comparison results are shown in Fig 16.

From Fig 16, the model designed here performs the best, and the worst is the CNN model. From the actual training results, the dance sequences generated by CNN cannot produce obvious dance actions. The dance actions synthesized by the AM model have too large span, and the upper body, especially the swing of the arms, is too large, which is inconsistent with the aesthetics of conventional dance actions. The variation amplitude and frequency of the dance actions of the SA model are too small. The actions of the dance sequences synthesized by the model reported here conform to conventional dance postures and have rich and varied actions, which have initially met the requirements of music performance action generation.

Table 3 shows the comparison of loss function values of these different cross domain sequence models under different look back values.

**Table 2. Loss function and visual effect change data of different modules of Loss.**

| Model | LSTM-1 | LSTM-3 | LSTM-5 | LSTM (nots = 36) | LSTM (nots = 128) | LSTM (nots = 256) |
|---|---|---|---|---|---|---|
| Loss | 0.0036 | 0.0028 | 0.0028 | 0.00433 | 0.00282 | 0.00279 |

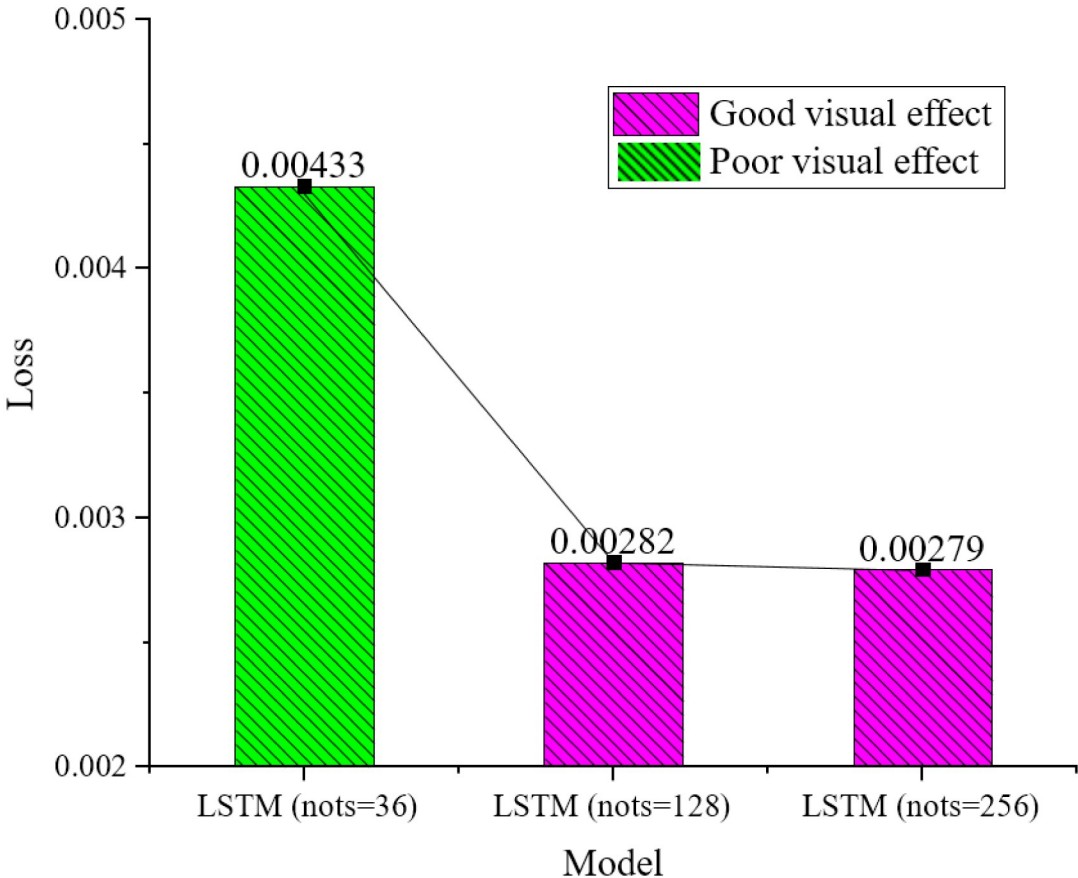

**Fig 14. Loss functions and visual effects with different numbers of nodes in the LSTM module.**

Table 3 suggests that as the lookback value increases, the loss values of different cross domain sequence models all increase to a certain extent. However, the improved model always maintains a loss value around 0.0022 and is lower than other models. This indicates that the improved model has less loss and better performance in establishing a connection between music and dance movements.

## 4. Discussion

The edge server structure and resource allocation strategy are analyzed. The results of this paper are compared with those of previous literature. Sun (2020) [21] studied the vocal teaching system of mobile edge computing. A system resource allocation method based on power iteration was proposed through the allocation and research of system teaching resources. The throughput of the unloading process was set as the objective function. It had practical reference value for promoting the heterogeneity of edge servers and the optimization of teaching resources. Hong et al. (2022) [22] researched the role of machine learning and artificial intelligence in music education for online games through the optimization of decision support systems. The results showed that innovative and complex methods based on artificial intelligence and machine learning were being used to improve music teaching. Hu et al. (2019) [23] studied artificial intelligence-assisted in-vehicle networks and proposed a strategy that integrated communication, caching, and computing to achieve cost-effectiveness of in-vehicle networks.

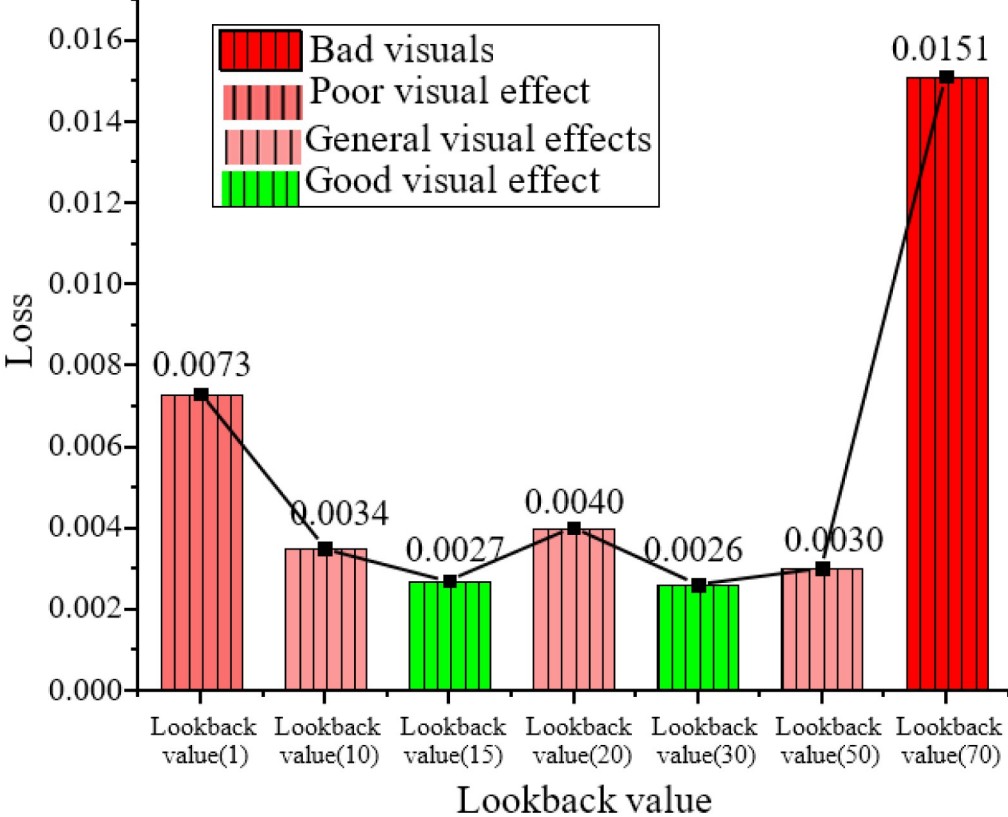

**Fig 15. Loss function values under different Lookback values.**

In addition, the statistical test results show that the proposed model structure based on edge computing and RNN has the highest accuracy of movement recognition. In summary, the proposed edge cache strategy can optimize the structure of the system, and the proposed edge server hardware structure can improve the network structure model. The model loss value of the system is greatly reduced, and the accuracy of extraction and recognition of music and dance movements is greatly improved.

## 5. Conclusion

This paper focuses on the problem of motion generation in music performances. The rhythm of music is characterized to realize the interaction between music and dance actions. In addition, a new network model is constructed based on the attention mechanism and the long and short-term RNN. Model training and prediction are carried out based on videos on Music&Dance2019. The results indicate that the loss function result is the smallest, and the video effect is the best when the number of layers of the LSTM module in the model is 3, the node value is 256, and the lookback value is 15. Besides, edge computing and long short-term RNNs are integrated into the optimization of music performance and intelligent assistance systems. The main contribution is to solve the problems of long access time and limited network bandwidth during multi-user access through edge server architecture and system resource allocation. Also, the efficiency of data collection is improved. The necessity of this paper is to provide a reference for the optimal design of music teaching system. The results indicate that the new

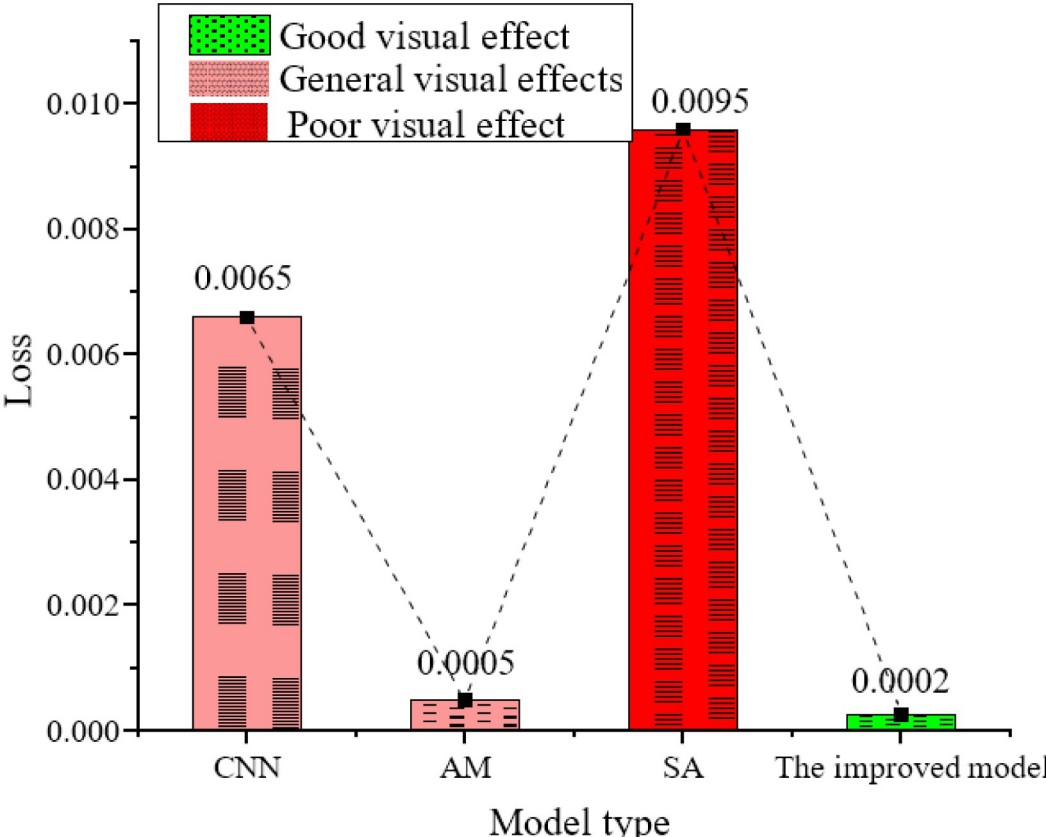

**Fig 16. Comparison of different performance action synthesis models.**

model can generate harmonious and rich performance action sequences based on ensuring stability. However, the research content is still limited and needs further improvement. The first is that the recognition of music beats is not ideal enough. The second is that the dance moves are still too monotonous and not smooth. The third is that an extensive database has not been established to achieve matching with music. The last point is that no suitable action character is designed, and the dance is too simplistic and abstract. It is hoped that in future studies and research, research in this area can be conducted to realize the generation of intelligent and humanized music performance actions.

**Table 3. Loss function and visual effect change data of different modules of Loss.**

| Lookback value | Model | | | |
|---|---|---|---|---|
| | CNN | AM | SA | The improved model |
| 10 | 0.0026 | 0.0028 | 0.0029 | 0.0021 |
| 20 | 0.0027 | 0.0029 | 0.0031 | 0.0022 |
| 30 | 0.0028 | 0.0031 | 0.0033 | 0.0026 |
| 40 | 0.0031 | 0.0035 | 0.0037 | 0.0022 |
| 50 | 0.0049 | 0.0052 | 0.0059 | 0.0025 |

## Supporting information

**S1 Data.**
(ZIP)

## Author Contributions

**Conceptualization:** Yi Wang.

**Investigation:** Yi Wang.

**Supervision:** Yi Wang.

**Visualization:** Yi Wang.

**Writing – original draft:** Yi Wang.

**Writing – review & editing:** Yi Wang.

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
