## [Decision Letter · Decision Letter 0]

24 Feb 2023

PONE-D-22-29932Intelligent Auxiliary System for Music Performance under Edge Computing and Long Short-Term Recurrent Neural NetworksPLOS ONE

Dear Dr. Wang,

Thank you for submitting your manuscript to PLOS ONE. After careful consideration, we feel that it has merit but does not fully meet PLOS ONE’s publication criteria as it currently stands. Therefore, we invite you to submit a revised version of the manuscript that addresses the points raised during the review process.

We look forward to receiving your revised manuscript.

Kind regards,

Muhammad Fazal Ijaz

Academic Editor

PLOS ONE

Journal Requirements:

3. PLOS requires an ORCID iD for the corresponding author in Editorial Manager on papers submitted after December 6th, 2016. Please ensure that you have an ORCID iD and that it is validated in Editorial Manager. To do this, go to ‘Update my Information’ (in the upper left-hand corner of the main menu), and click on the Fetch/Validate link next to the ORCID field. This will take you to the ORCID site and allow you to create a new iD or authenticate a pre-existing iD in Editorial Manager. Please see the following video for instructions on linking an ORCID iD to your Editorial Manager account: https://www.youtube.com/watch?v=_xcclfuvtxQ.

4. We note that Figure10 includes an image of a [patient / participant / in the study].

Reviewers' comments:

Reviewer's Responses to Questions

**Comments to the Author**

1. Is the manuscript technically sound, and do the data support the conclusions?

Reviewer #1: Partly

Reviewer #2: Yes

2. Has the statistical analysis been performed appropriately and rigorously? 

Reviewer #1: Yes

Reviewer #2: Yes

3. Have the authors made all data underlying the findings in their manuscript fully available?

Reviewer #1: Yes

Reviewer #2: Yes

4. Is the manuscript presented in an intelligible fashion and written in standard English?

Reviewer #1: No

Reviewer #2: Yes

5. Review Comments to the Author

Reviewer #1: The overall impression of the technical contribution of the current study is reasonable. However, the Authors may consider making necessary amendments to the manuscript for better comprehensibility of the study.

1. The abstract must be re-written, focusing on the technical aspects of the proposed model, the main experimental results, and the metrics used in the evaluation. Briefly discuss how the proposed model is superior.

2. Authors are recommended to re-check the keywords.

3. Provide adequate citations for the contenst of Introduction, like line 39 & 40.

4. The contribution of the current study must be briefly discussed as bullet points in the introduction. And motivation must also be discussed in the manuscript.

5. The overall organization of the manuscript is not discussed anywhere in the manuscript. Please add the same in the introduction section of the manuscript.

6. Introduction section must discuss the technical gaps associated with the current problem.

7. The literature section may be further improvised by incorporating some of the relevant studies like https://doi.org/10.3390/s21082852 and https://doi.org/10.1371/journal.pone.0262708

8. Authors may provide the architecture/block diagram of the proposed model for better comprehensibility of the proposed model concerning various aspects of the proposed model.

9.The important details, like the input/tensor/kernel size, must be discussed, and whether authors have used Stride 1 or Stride, 2 must be presented. What type of activation function is being used in the current study.(Add if possible)

10. For how many epochs does the proposed model execute. what is the initial learning rate, and after how many epochs does the model's learning rate saturated.(Must add details).

11. Authors must provide the details of hyper parameters like training loss, testing loss, training accuracy and testing accuracy. check and include reference https://doi.org/10.3390/s22082988(Add graphs if possible)

12. More comparative analysis with state-of-art models is desired. (Must do more comparative analysis).

13. By considering the current form of the conclusion section, it is hard to understand by PLOS Journal readers. It should be extended with new sentences about the necessity and contributions of the study by considering the authors' opinions about the experimental results derived from some other well-known objective evaluation values if it is possible. (Add if possible)

14. Authors should use more alternative models as the benchmarking models, authors should also conduct some statistical tests to ensure the superiority of the proposed approach, i.e., how could authors ensure that their results are superior to others? Meanwhile, the authors also have to provide some insightful discussion of the results.(Add if possible)

Reviewer #2: 1) The content is innovative, and the results are encouraging. So, the manuscript can be accepted.

2) In figures 13, 15, and 16, loss values should be written above each column.

3) Parameter setting table should be added to section result. In that table, number of layers, units, epoch, batch size etc. should be described.

4) The comparison results of different models should be presented in a table.

6. PLOS authors have the option to publish the peer review history of their article (what does this mean?). If published, this will include your full peer review and any attached files.

Reviewer #1: No

Reviewer #2: No

---

## [Author Response · Author response to Decision Letter 0]

29 Mar 2023

Reviewer #1: The overall impression of the technical contribution of the current study is reasonable. However, the Authors may consider making necessary amendments to the manuscript for better comprehensibility of the study.

1. The abstract must be re-written, focusing on the technical aspects of the proposed model, the main experimental results, and the metrics used in the evaluation. Briefly discuss how the proposed model is superior.

Reply: Thank you for your valuable advice. In terms of technology, the model adopts the image description generation model with attention mechanism, and combines the RNN abstract structure without considering recursion to optimize the abstract network structure of RNN-LSTM. The measurement standard of experimental results and evaluation is the model loss function value. The superiority of the suggested model is mainly reflected in the high accuracy and low consumption rate of dance movement recognition. The main technical advantages and evaluation criteria of the proposed model have been supplemented in detail in the abstract.

2. Authors are recommended to re-check the keywords.

Reply: Thank you for your comments. The contents of keywords have been rechecked, and keywords such as "music, performance action" have been deleted, and appropriate keywords that are consistent with the research topic have been added.

3. Provide adequate citations for the contenst of Introduction, like line 39 & 40.

Reply: Thank you for your valuable advice. More documents have been added to the introduction for reference. The serial number of the supplementary documents is [1,2]. Thank you again for your valuable advice.

4. The contribution of the current study must be briefly discussed as bullet points in the introduction. And motivation must also be discussed in the manuscript.

Reply: Thank you for your valuable advice. The main contribution of the research is to analyze and optimize the relationship between music and dance performance by using the attention mechanism model. The main motivation of the research is to design and apply the intelligent assistant system for music performance through edge computing and long-term and short-term recurrent neural networks, so as to improve the performance level of music and dance movements. The contribution and main motivation of the research have been supplemented and discussed in detail in the introduction.

5. The overall organization of the manuscript is not discussed anywhere in the manuscript. Please add the same in the introduction section of the manuscript.

Reply: Thank you for your comments. The first section of the manuscript is background introduction, the second section is explanation of research methods, the third section is analysis and discussion of research results, and the fourth section is summary of research conclusions. The overall structure of the manuscript has been supplemented in detail in the last paragraph of the introduction section.

6. Introduction section must discuss the technical gaps associated with the current problem.

Reply: Thank you for your opinion. There is a certain gap between the current technology and the expected recognition accuracy of the model. The current model has a low level of musical intelligence assistance, and it cannot effectively combine performance movements with musical beats. The detailed discussion on the related technical gap has been supplemented in the introduction section of the research.

7. The literature section may be further improvised by incorporating some of the relevant studies like https://doi.org/10.3390/s21082852 and https://doi.org/10.1371/journal.pone.0262708

Reply: Thank you for your valuable comments. The studies you mentioned have been supplemented in detail in the literature review section, and the number of the added studies is [5] and [6]. Thank you again for your valuable comments.

8. Authors may provide the architecture/block diagram of the proposed model for better comprehensibility of the proposed model concerning various aspects of the proposed model.

Reply: Thank you for your valuable comments. The architecture diagram of the proposed model has been supplemented in detail in Figure 7. Thank you again for your comments.

9.The important details, like the input/tensor/kernel size, must be discussed, and whether authors have used Stride 1 or Stride, 2 must be presented. What type of activation function is being used in the current study.(Add if possible)

Reply: Thank you for your valuable comments. The Input of the study is 50, Tensor is 100, and Kernel size is 2*2. Stride 2 is used in the study, and the type of activation function is Sigmoid. More important parameter details have been supplemented in Table 2. Thank you again for your comments.

10. For how many epochs does the proposed model execute. what is the initial learning rate, and after how many epochs does the model's learning rate saturated.(Must add details).

Reply: Thank you for your valuable advice. The proposed model has carried out 500 iterations, with an initial learning rate of 0.005, and the learning rate of the model is saturated after 400 iterations. More parameter information about the model has been supplemented in Section 3 and Table 1.

11. Authors must provide the details of hyper parameters like training loss, testing loss, training accuracy and testing accuracy. check and include reference https://doi.org/10.3390/s22082988(Add graphs if possible)

Reply: Thank you for your valuable comments. The training loss is 0.005, the test loss is 0.004, the training accuracy is 0.74, and the testing accuracy is 0.76. More details of the hyperparameter have been supplemented in Table 1. In addition, the literature provided has been referenced, and the supplementary literature is numbered [20]. Thank you again for your valuable advice.

12. More comparative analysis with state-of-art models is desired. (Must do more comparative analysis).

Reply: Thank you for your valuable advice. The research results of this paper have been analyzed with the most advanced model results before. The proposed edge caching strategy can optimize the structure of the system, and the proposed hardware structure of the edge server can improve the network structure model. The model loss value of the system is greatly reduced, and the accuracy of the extraction and recognition of music and dance movements is greatly improved. More comparative analysis results have been supplemented in detail in the discussion section of the study.

13. By considering the current form of the conclusion section, it is hard to understand by PLOS Journal readers. It should be extended with new sentences about the necessity and contributions of the study by considering the authors' opinions about the experimental results derived from some other well-known objective evaluation values if it is possible. (Add if possible)

Reply: Thank you for your valuable opinions. The objective evaluation and analysis results of the research results are summarized as follows. The main contribution of the research is to solve the problems of long access time and limited network bandwidth when multi-users access through the edge server architecture and system resource allocation, and improve the efficiency of data collection. The necessity of the research is to provide reference for the optimal design of music teaching system. The necessity and contribution of the study have been supplemented in detail in the conclusion section of the study.

14. Authors should use more alternative models as the benchmarking models, authors should also conduct some statistical tests to ensure the superiority of the proposed approach, i.e., how could authors ensure that their results are superior to others? Meanwhile, the authors also have to provide some insightful discussion of the results.(Add if possible)

Reply: Thank you for your valuable advice. The statistical test results show that the model structure based on edge computing and recurrent neural network proposed in the study has the highest accuracy in action recognition. More discussion about the research results has been supplemented in detail in Section 4 of the study.

Reviewer #2: 

1) The content is innovative, and the results are encouraging. So, the manuscript can be accepted.

Reply: Thank you for your valuable comments, and sincerely thank you for your contribution in the process of article review.

2) In figures 13, 15, and 16, loss values should be written above each column.

Reply: Thank you for your valuable comments. Figures 13, 15 and 16 have been revised, and the loss function values have been supplemented at the top of each column. Thank you again for your valuable comments.

3) Parameter setting table should be added to section result. In that table, number of layers, units, epoch, batch size etc. should be described.

Reply: Thank you for your valuable comments. The parameter results such as number of layers, units, epoch, batch size and threshold value iterations have been described in Table 1. Thank you again for your valuable comments.

4) The comparison results of different models should be presented in a table.

Reply: Thank you for your comments. The comparison result of loss value of LSTM-1 is 0.0036, that of LSTM-3 and LSTM-5 is 0.0028, that of LSTM (nots=36) is 0.00433, and that of LSTM (nots=128) is 0.00282. The performance comparison results of different models have been supplemented in detail in Table 2.

1. We note that Figure10 includes an image of a [patient / participant / in the study].

Please provide your response in "Author Comments" box.

Reply: Thank you for your valuable advice. The content of Figure 10 has been modified, and the image of the participant in Figure 10 has been deleted and replaced by a simulated robot. Thank you again for your audit opinion.

---

## [Decision Letter · Decision Letter 1]

11 Apr 2023

PONE-D-22-29932R1Intelligent Auxiliary System for Music Performance under Edge Computing and Long Short-Term Recurrent Neural NetworksPLOS ONE

Dear Dr. Wang,

Thank you for submitting your manuscript to PLOS ONE. After careful consideration, we feel that it has merit but does not fully meet PLOS ONE’s publication criteria as it currently stands. Therefore, we invite you to submit a revised version of the manuscript that addresses the points raised during the review process.

We look forward to receiving your revised manuscript.

Kind regards,

Muhammad Fazal Ijaz

Academic Editor

PLOS ONE

Journal Requirements:

Reviewers' comments:

Reviewer's Responses to Questions

**Comments to the Author**

1. If the authors have adequately addressed your comments raised in a previous round of review and you feel that this manuscript is now acceptable for publication, you may indicate that here to bypass the “Comments to the Author” section, enter your conflict of interest statement in the “Confidential to Editor” section, and submit your "Accept" recommendation.

Reviewer #1: All comments have been addressed

Reviewer #2: (No Response)

2. Is the manuscript technically sound, and do the data support the conclusions?

Reviewer #1: Yes

Reviewer #2: Yes

3. Has the statistical analysis been performed appropriately and rigorously? 

Reviewer #1: Yes

Reviewer #2: Yes

4. Have the authors made all data underlying the findings in their manuscript fully available?

Reviewer #1: Yes

Reviewer #2: No

5. Is the manuscript presented in an intelligible fashion and written in standard English?

Reviewer #1: Yes

Reviewer #2: No

6. Review Comments to the Author

Reviewer #1: Authors have done necessary amendments in line with the recommendation of the reviewer. The manuscript in the current form may considered for the further phase of the editorial procedure.

Reviewer #2: 1) Parameter setting table is not suitable. Parameter name and values should be written in columns, in the first column parameter name and in the second one parameter values.

2) Author just added one table for LSTM. For the other methods such as lookback value and comparison of different synthesis models do not have table. They should also add tables to mentioned methods and then explain the result of each table.

7. PLOS authors have the option to publish the peer review history of their article (what does this mean?). If published, this will include your full peer review and any attached files.

Reviewer #1: No

Reviewer #2: No

---

## [Author Response · Author response to Decision Letter 1]

13 Apr 2023

Reviewer #1: Authors have done necessary amendments in line with the recommendation of the reviewer. The manuscript in the current form may considered for the further phase of the editorial procedure.

Reply: Okay, thank you for your affirmation after careful reading. I am glad to enter the next stage and look forward to your reply.

Reviewer #2: 1) Parameter setting table is not suitable. Parameter name and values should be written in columns, in the first column parameter name and in the second one parameter values.

Reply: Okay, thank you for your feedback after carefully reading. I have adjusted the format of the parameter setting table, with the first column being the parameter name and the second column being the parameter value. Please refer to Table 1.

2) Author just added one table for LSTM. For the other methods such as lookback value and comparison of different synthesis models do not have table. They should also add tables to mentioned methods and then explain the result of each table.

Reply: OK, thank you for your comments after careful reading. I have added Table 3 below Section 3.2 to compare the loss function values of different cross domain sequence models under different look back values, to further reflect the advantages of the model proposed in this study.

---

## [Decision Letter · Decision Letter 2]

25 Apr 2023

Intelligent Auxiliary System for Music Performance under Edge Computing and Long Short-Term Recurrent Neural Networks

PONE-D-22-29932R2

Dear Dr. Wang,

We’re pleased to inform you that your manuscript has been judged scientifically suitable for publication and will be formally accepted for publication once it meets all outstanding technical requirements.

Kind regards,

Muhammad Fazal Ijaz

Academic Editor

PLOS ONE

Additional Editor Comments (optional):

Reviewers' comments:

Reviewer's Responses to Questions

**Comments to the Author**

1. If the authors have adequately addressed your comments raised in a previous round of review and you feel that this manuscript is now acceptable for publication, you may indicate that here to bypass the “Comments to the Author” section, enter your conflict of interest statement in the “Confidential to Editor” section, and submit your "Accept" recommendation.

Reviewer #1: All comments have been addressed

Reviewer #2: All comments have been addressed

2. Is the manuscript technically sound, and do the data support the conclusions?

Reviewer #1: Yes

Reviewer #2: Yes

3. Has the statistical analysis been performed appropriately and rigorously? 

Reviewer #1: Yes

Reviewer #2: Yes

4. Have the authors made all data underlying the findings in their manuscript fully available?

Reviewer #1: Yes

Reviewer #2: Yes

5. Is the manuscript presented in an intelligible fashion and written in standard English?

Reviewer #1: Yes

Reviewer #2: Yes

6. Review Comments to the Author

Reviewer #1: The authors have addressed all the recommendations of the reviewers in a reasonable manner, manuscript in the current from may be considered for the further phase of editorial process.

Reviewer #2: Author has done all amendments suggested by reviewers. I confirm that this paper is a suitable one for publication.

7. PLOS authors have the option to publish the peer review history of their article (what does this mean?). If published, this will include your full peer review and any attached files.

Reviewer #1: No

Reviewer #2: No

---

## [Editor Report · Acceptance letter]

28 Apr 2023

PONE-D-22-29932R2 

Intelligent Auxiliary System for Music Performance under Edge Computing and Long Short-Term Recurrent Neural Networks 

Dear Dr. Wang:

I'm pleased to inform you that your manuscript has been deemed suitable for publication in PLOS ONE. Congratulations! Your manuscript is now with our production department. 

Kind regards, 

on behalf of

Dr. Muhammad Fazal Ijaz 

Academic Editor

PLOS ONE